# NeRF Compression via Transform Coding

## Abstract

Neural Radiance Fields (NeRFs) have emerged as powerful tools for capturing detailed 3D scenes through continuous volumetric representations. Recent NeRFs utilize feature grids to improve rendering quality and speed; however, these representations introduce significant storage overhead. This paper presents a novel method for efficiently compressing a grid-based NeRF model. Our approach is based on the non-linear transform coding paradigm, where we compress the model's feature grids using end-to-end optimized neural compression. Since these neural compressors are overfitted to individual scenes, we develop lightweight decoders and encoder-free compression. To exploit the spatial inhomogeneity of the latent feature grids, we introduce an importance-weighted rate-distortion objective and a sparse entropy model using a masking mechanism. Our experimental results validate that our proposed method surpasses existing works in terms of grid-based NeRF compression efficacy and reconstruction quality.

## 1 Introduction

Over the past few years, the field of 3D scene modeling and reconstruction has been revolutionized by the advent of Neural Radiance Fields (NeRF) methodologies (Mildenhall et al., 2021; Zhang et al., 2020; Barron et al., 2021). NeRF offers a sophisticated method for 3D reconstruction, with the ability to synthesize novel viewpoints from limited 2D data. Yet, for all its merits, the original NeRF model requires millions of MLP queries, which causes a slow training and rendering time.

To address the efficiency concerns surrounding the traditional NeRF, recent advancements have transitioned to the integration of an explicit grid representation (Yu et al., 2021; Sun et al., 2022; Fridovich-Keil et al., 2022; Chen et al., 2022; Fridovich-Keil et al., 2023; Chan et al., 2022). While accelerating training and rendering processes significantly, this change also poses a new challenge: the storage costs for saving the explicit grid NeRF representation significantly increase. This problem is crucial, especially in real-world applications where storage and transmission are critical constraints.

Our work seeks to significantly reduce the storage costs of NeRFs. Inspired by the neural image compression methodology (Yang et al., 2023), we apply non-linear transform coding techniques (Ballé et al., 2020) to compress the explicit grid NeRF representation efficiently. However, we sidestep the conventional auto-encoder approach in favor of an iterative inference framework, in which we jointly optimize the latent code along with a lightweight decoder. We further take account of the NeRF grid importance scores while reconstructing the scene to boost the efficiency of our compression model. Lastly, we propose a novel entropy model that masks out uninformative feature grid points. Utilizing a rate-distortion objective, we are flexible in choosing from various compression levels. Our proposed approach departs from previous works on compressing explicit grid NeRF representation (Li et al., 2023a;b; Deng & Tartaglione, 2023), which are based on voxel pruning and/or vector quantization (Gray, 1984) to omit the least importance voxels and reduce the storage size.

To show the effectiveness of our proposed method, we perform extensive experiments on four different datasets. Our results show that our model is capable of compressing diverse NeRF scenes to a much smaller size and outperforms previous works.

## 2 BACKGROUND

### 2.1 NEURAL RADIANCE FIELDS

Neural radiance fields (Mildenhall et al., 2021), or NeRF, mark a paradigm shift in 3D scene representation using deep neural networks. Instead of relying on traditional discrete representations like point clouds or meshes, NeRF models a scene as a continuous volumetric function $F : (\mathbf{x}, \mathbf{d}) \rightarrow (\mathbf{c}, \sigma)$, in which an input comprising of spatial coordinates $\mathbf{x}$ and a viewing direction $\mathbf{d}$ is mapped to an output representing color $\mathbf{c}$ and volume density $\sigma$. The expected color $\hat{C}(\mathbf{r})$ for the ray $\mathbf{r}$ can be calculated by sampling $N$ points along the ray:

$$\hat{C}(\mathbf{r}) = \sum_{i=1}^{N} T_i \cdot \alpha_i \cdot \mathbf{c}_i; \quad \alpha_i = 1 - \exp(-\sigma_i \delta_i), \quad T_i = \prod_{j=1}^{i}(1 - \alpha_j) \tag{1}$$

where $\delta_i$ is the distance between adjacent sampled points, $\alpha_i$ is the probability of termination at $i$-th point, and $T_i$ is the accumulated transmittance along the ray up to $i$-th point. NeRF is then trained to minimize total squared error loss between the rendered and true pixel colors.

$$\mathcal{L}_{render} = \sum_{\mathbf{r}} ||\hat{C}(\mathbf{r}) - C(\mathbf{r})||_2^2 \tag{2}$$

Despite NeRF's ability to provide intricate scene details with a relatively compact neural network, the computational demand remains a significant constraint. The evaluation over the volume often requires thousands of network evaluations per pixel. To reduce training and inference time, recent research has employed explicit grid structure into NeRF. More specifically, they introduce voxel grids (Sun et al., 2022; Fridovich-Keil et al., 2022) or decomposed feature planes (Chen et al., 2022; Fridovich-Keil et al., 2023; Chan et al., 2022) into the model, and query point features via trilinear or bilinear interpolation. While this notably speeds up training and inference, it does come at the expense of greater storage needs from saving the feature grids.

### 2.2 NEURAL COMPRESSION

Neural compression utilizes neural networks to perform data compression tasks. Traditional compression algorithms are handcrafted and specifically tailored to the characteristics of the data they compress, such as JPEG (Wallace, 1991) for images or MP3 for audio. In contrast, neural compression seeks to learn efficient data representations directly from the data, exemplified by the nonlinear transform coding paradigm (Ballé et al., 2020).

Existing lossy neural compression methods (Ballé et al., 2016; 2018; Minnen et al., 2018; Cheng et al., 2020) often leverages an auto-encoder structure (Kingma & Welling, 2013). An encoder $E$ maps a given data $\mathbf{X}$ to a continuous latent representation $\mathbf{Z}$. This continuous latent representation $\mathbf{Z}$ is then quantized to integers by $Q$: $\hat{\mathbf{Z}} = Q(\mathbf{Z})$. An entropy model $P$ is used to transmit $\hat{\mathbf{Z}}$ losslessly. The decoder $D$ receives the quantized latent code $\hat{\mathbf{Z}}$, and reconstruct the original data $\hat{\mathbf{X}} = D(\hat{\mathbf{Z}})$. We train the encoder $E$, the decoder $D$ and the entropy model $P$ jointly using a rate-distortion objective:

$$\mathcal{L}(E, D, P) = \mathbb{E}_{\mathbf{X} \sim p(\mathbf{X})} d(\mathbf{X}, D(Q(E(\mathbf{X})))) - \lambda \log_2 P(Q(E(\mathbf{X}))) \tag{3}$$

where the first term is the distortion loss that measures the reconstruction quality, the second term is the rate loss that measures the expected code length, and $\lambda$ is a hyperparameter to balance between the two loss terms. At training time, the quantizer $Q$ is typically replaced with uniform noise adding (Ballé et al., 2016) to enable end-to-end training. See Yang et al. (2023) for a detailed review on neural compression.

## 3 METHOD

In this section, we describe our method for grid-based NeRF compression. Our primary focus is on the compression of the TensoRF-VM model (Chen et al., 2022), characterized by its decomposed 2D feature planes structure (Kolda & Bader, 2009). We select TensoRF-VM because of its proficient 3D scene modeling capabilities, often outperforms alternative methods like Plenoxels (Fridovich-Keil et al., 2022) or DVGO (Sun et al., 2022). Nevertheless, it's worth mentioning that our method has the potential to be applied to other grid-based NeRF architectures.

**Problem setting.** We have a TensoRF-VM model that was pre-trained for a single scene, and our task is to reduce its size through compression, while maintaining its reconstruction quality. We assume that we have access to the training dataset comprising view images at compressing time.

**Notation.** The three feature planes (or matrix components) of TensoRF-VM are denoted by $\{\mathbf{P}_i\}_{i=1}^3$, in which subscript $i$ signifies the index of the planes and each $\mathbf{P}_i \in \mathbb{R}^{C_i \times H_i \times W_i}$. In practice, TensoRF-VM has two types of feature planes: density planes for calculating densities and appearance planes for calculating colors. Our $\{\mathbf{P}_i\}_{i=1}^3$ can be thought of as a channel-wise concatenation of those two types. The vector components are not considered in our compression, hence, are not represented in our notation. For indexing a specific spatial grid location $j$ in the feature plane $i$, we employ a superscript, represented as $\mathbf{P}_i^j$.

## 3.1 COMPRESSING THE FEATURE PLANES

Given a trained TensoRF-VM model, most storage is used to store the feature grids. To illustrate this, we analyze a trained model for the Lego scene from Synthetic-NeRF dataset (Mildenhall et al., 2021). In this model, the 2D feature planes takes 67.61 MB, while the other components such as the rendering MLP, the rendering mask, and the vector components takes only 1.21 MB. Given this disparity, a straightforward approach to compress the TensoRF-VM model would be to employ neural compression specifically on the feature planes.

In details, we can define an encoder $E$ that takes the there feature planes $\{\mathbf{P}_i\}_{i=1}^3$ and embeds them to latent codes $\{\mathbf{Z}_i\}_{i=1}^3$, in which the $\mathbf{Z}_i$ may have lower resolution than $\mathbf{P}_i$. The latent codes are quantized to $\{\hat{\mathbf{Z}}_i\}_{i=1}^3$ and compressed with entropy coding using an entropy model $P$. At rendering time, we decompress the quantized latent codes $\{\hat{\mathbf{Z}}_i\}_{i=1}^3$ and forward them to the decoder $D$ to reconstruct the three feature planes $\{\hat{\mathbf{P}}_i\}_{i=1}^3$. We then use $\{\hat{\mathbf{P}}_i\}_{i=1}^3$ to query sampling point features and render the scene. The compressed NeRF model includes the compressed latent codes, the decoder, the entropy model and the other components.

It is crucial to highlight that we only need to reconstruct the three feature planes once, and all subsequent querying operations for the sampling points are executed on these reconstructed planes. Thus, the decompression process only adds minimal overhead to the overall rendering procedure.

**Removing the encoder.** In nonlinear transform coding, the encoder is used to obtain the latent code of a new data point via amortized inference (Kingma & Welling, 2013; Gershman & Goodman, 2014). This is essential for compressing a new data point quickly in a single network pass. Nonetheless, in the case of TensoRF-VM compression, our primary objective is to compress merely the three feature planes, and our decoder is overfitted to a single scene. Moreover, using an encoder for amortized inference leads to an irreducible amortization gap in optimization (Cremer et al., 2018; Marino et al., 2018), which has been shown to degrade compression performance (Campos et al., 2019; Yang et al., 2020).

To tackle the aforementioned problems, we propose to remove the encoder from our compression pipeline and directly learn the three latent codes $\{\mathbf{Z}_i\}_{i=1}^3$ from scratch. More specifically, we initialize the $\{\mathbf{Z}_i\}_{i=1}^3$ as zeros tensor, and jointly optimize $\{\mathbf{Z}_i\}_{i=1}^3$ with the decoder $D$ and the probability model $P$.

**Architecture design.** Since we must transmit the decoder $D$ along with the latent code $\{\mathbf{Z}_i\}_{i=1}^3$ to decompress the scene, it's essential for the decoder to be lightweight. Yang & Mandt (2023) has shown the effective of a lightweight decoder for neural compression. We found that a two-layer transposed convolutional neural network with SELU activation Klambauer et al. (2017) is effective for our needs.

## 3.2 IMPORTANCE-WEIGHTED TRAINING LOSS

Our model is trained end-to-end with a rate-distortion loss. The rate loss is defined as the log-likelihood of the entropy model $P$, and it ensures that the compressed feature planes remain compact. For the distortion loss, we discover that using only the NeRF rendering loss is not sufficient, and we also need to use a L2 feature planes reconstruction loss for a good rendering quality.

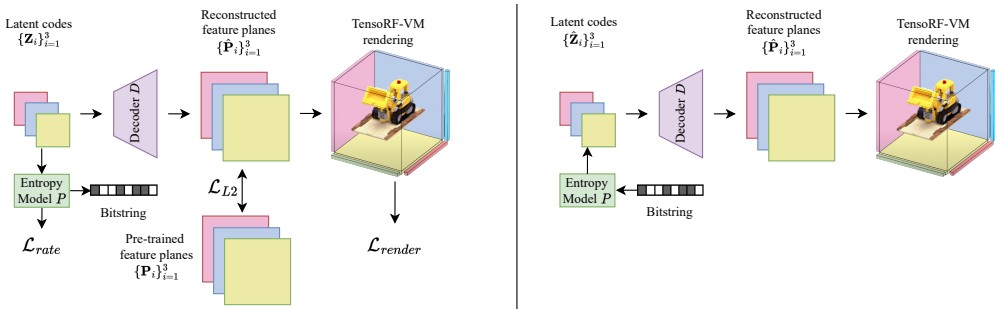

Figure 1: **Overview of our model.** At training time (left), we learn the three latent codes $\{\mathbf{Z}_i\}_{i=1}^3$ to reconstruct the three frozen feature planes $\{\mathbf{P}_i\}_{i=1}^3$. The reconstructed feature planes $\{\hat{\mathbf{P}}_i\}_{i=1}^3$. are used to render the scene and calculate the rendering loss. The entropy model $P$ is used to calculate the rate loss and compress the latent codes to bitstring. At rendering time (right), we use $P$ to decompress the bitstring to latent codes $\{\hat{\mathbf{Z}}_i\}_{i=1}^3$ and then reconstruct the feature planes $\{\hat{\mathbf{P}}_i\}_{i=1}^3$.

However, reconstructing the entire feature planes is not the most efficient approach for compression. Prior research (Li et al., 2023a;b; Deng & Tartaglione, 2023) has illustrated that these feature grids possess significant redundancy, and could be pruned to decrease the size of the model. Consequently, if we were to reconstruct every single grid location, it would inevitably lead to additional storage cost.

To address this issue, we suggest computing the rendering importance score, which then acts as a weight for the L2 feature planes reconstruction loss. With this approach, our model is guided to reconstruct only important grid locations, while ignoring the less crucial ones, ensuring a more optimized and effective compression representation.

For each feature plane $\mathbf{P}_i \in \mathbb{R}^{C_i \times W_i \times H_i}$, we define $\mathbf{W}_i \in \mathbb{R}^{W_i \times H_i}$ as the corresponding weight map, and this weight is shared across $C$ feature channels. To determine $\{\mathbf{W}_i\}_{i=1}^3$, we first follow Li et al. (2023a;b) to calculate the rendering importance score $\{\mathbf{I}_i\}_{i=1}^3$ as follows: For each grid location $\mathbf{P}_i^j$, its importance $\mathbf{I}_i^j$ can be determined by accumulating the importance scores of every sampling points $\mathbf{x}_k$ that contribute to it:

$$\mathbf{I}_i^j = \sum_{k \in \mathcal{N}_j} \omega_k^j \cdot T_k \cdot \alpha_k \tag{4}$$

where $T_k$ and $\alpha_k$ are defined in Section 2.1, $\mathcal{N}_j$ denotes the set of sampling points that falls within the neighborhood of $\mathbf{v}_j$, and $\omega_k^j$ denotes the bilinear interpolation weight of $\mathbf{x}_k$ that corresponds to the feature $\mathbf{P}_i^j$.

We then apply a log-function to the importance maps $\{\mathbf{I}_i\}_{i=1}^3$, and then normalize them to the range of $[0,1]$ to get the weights $\{\mathbf{W}_i\}_{i=1}^3$:

$$\mathbf{W}_i = \text{normalize}(\log(\mathbf{I}_i + \epsilon)) \tag{5}$$

in which $\epsilon$ is a small number to ensure that the log function is well-defined.

Our final loss function is:

$$\mathcal{L} = \mathcal{L}_{render}(\{\hat{\mathbf{P}}_i\}_{i=1}^3) + \sum_{i=1}^3 \left( ||\mathbf{P}_i - \hat{\mathbf{P}}_i||_2^2 \otimes \mathbf{W}_i - \lambda \log_2 P(\hat{\mathbf{Z}}_i) \right) \tag{6}$$

We train the decoder $D$, the entropy model $P$, the planes $\{\mathbf{Z}_i\}_{i=1}^3$, and also fine-tune the rendering MLP and the vector components of TensoRF-VM model with this loss function.

### 3.3 MASKED ENTROPY MODEL

Applying neural compression for TensoRF-VM enables us to use a wide range of different entropy models. In this section, we design a simple but effective entropy models that works well for TensoRF-VM compression.

Theoretically, using a learned entropy model $P$ to compress the three feature planes enables the entropy coder to devise an optimal coding strategy, aiming to reduce redundancy. In our empirical assessments, we observed that a predominant portion of the learned latent code is zero, especially in the background. This observation might be attributed to our choice of initializing the latent codes as zero tensors. Consequently, even though the entropy coder can efficiently map the zeros to shorter bit strings, there remains the overhead of transmitting them.

In order to avoid compressing redundant information, we propose to incorporate binary masks $\{\mathbf{M}_i\}_{i=1}^3$ into our entropy model $P$. The model $P$ compresses grid features $\mathbf{P}_i^j$ only when $\mathbf{M}_i^j = 1$, allowing selective compression of specific features and avoiding others. Those masks are learnable, and can be treated as additional parameters of $P$.

In details, we design $P$ to be a fully factorized probability distribution from Ballé et al. (2016) and Ballé et al. (2018), in which every grid location is independent and identically distributed by a non-parametric distribution $p_\theta$ with learnable $\theta$. For each latent code $\hat{\mathbf{Z}}_i$ to be compressed, we establish a corresponding binary mask $\mathbf{M}_i$ that has the same spatial size and shared across features channel. The joint probability distribution $P$ is then factorized across spatial locations $j$ as:

$$P_{\mathbf{M}_i}(\hat{\mathbf{Z}}_i) = \prod_j p(\hat{\mathbf{Z}}_i^j | \mathbf{M}_i^j)$$

$$p(\hat{\mathbf{Z}}_i^j | \mathbf{M}_i^j = 0) = \delta(\hat{\mathbf{Z}}_i^j = 0); \quad p(\hat{\mathbf{Z}}_i^j | \mathbf{M}_i^j = 1) = p_\theta(\hat{\mathbf{Z}}_i^j) \tag{7}$$

Here, $\delta$ represents the Dirac delta distribution, which allocates a probability of 1 to $\hat{\mathbf{Z}}_i^j = 0$ and 0 otherwise. This implies that if $\mathbf{M}_i^j = 0$, then we designate $\hat{\mathbf{Z}}_i^j = 0$. Thus, the input to the decoder $D$ can be calculated as $\hat{\mathbf{Z}}_i \otimes \mathbf{M}_i$, and the reconstructed planes are:

$$\hat{\mathbf{P}}_i = D(\hat{\mathbf{Z}}_i \otimes \mathbf{M}_i) \tag{8}$$

However, since the masks $\mathbf{M}_i$ are binary, they cannot be learned directly. To address this, we turn to the Gumbel-Softmax trick (Jang et al., 2016) to facilitate the learning of $\mathbf{M}_i$. For each $\mathbf{M}_i$, we define the 2-classes probability denoted by $\pi_{\mathbf{M}_i}^0$ and $\pi_{\mathbf{M}_i}^1$ that assign the probabilities of the mask equal to 0 and 1. At training time, we sample $\mathbf{M}_i$ by straight through Gumbel-Softmax estimator (Bengio et al., 2013; Jang et al., 2016):

$$\mathbf{M}_i = \underset{j \in \{0,1\}}{\arg\max}(g_j + \log \pi_{\mathbf{M}_i}^j) \tag{9}$$

in which $g_j$ are i.i.d samples drawn from Gumbel$(0, 1)$. The straight through Gumbel-Softmax estimator allows us to calculate the gradients of $\pi_{\mathbf{M}_i}^j$. We then optimize the mask probabilities $\pi_{\mathbf{M}_i}^j$ following the rate-distortion loss:

$$\mathcal{L} = \mathcal{L}_{render}(\{\hat{\mathbf{P}}_i\}_{i=1}^3) + \sum_{i=1}^3 \left( ||\mathbf{P}_i - \hat{\mathbf{P}}_i||_2^2 \otimes \mathbf{W}_i - \lambda \log_2 P_{\mathbf{M}_i}(\hat{\mathbf{Z}}_i) \right) \tag{10}$$

where $\hat{\mathbf{P}}_i$ is calculated with Equation 8. In practice, we use an annealing softmax temperature $\tau$ that decays from 10 to 0.1 to calculate the softmax gradients. After training, the masks $\mathbf{M}_i$ are sent as additional parameters of the entropy model $P$.

## 4 EXPERIMENTS

### 4.1 EXPERIMENT SETTING

**Datasets.** We perform our experiments on 4 datasets:

- Synthetic-NeRF (Mildenhall et al., 2021): This dataset contains 8 scenes at resolution $800 \times 800$ rendered by Blender. Each scene contains 100 training views and 200 testing views.
- Synthetic-NSVF (Liu et al., 2020): This dataset also contains 8 rendered scenes at resolution $800 \times 800$. However Synthetic-NSVF contains more complex geometry and lightning effects compared to Synthetic-NeRF.

- LLFF (Mildenhall et al., 2019): LLFF contains 8 real-world scenes made of realistic and forward-facing images with non empty background. We use the resolution $1008 \times 756$.

- Tanks and Temples (Knapitsch et al., 2017): We use 5 real-world scenes: *Barn, Caterpillar, Family, Ignatus, Truck* from the Tanks and Temples dataset to experiment with. They have the resolution of $1920 \times 1080$.

To perform compression experiments, we initially train a TensoRF-VM model for every scene within those datasets. We use the default TensoRF-VM 192 hyperparameters, as detailed in the paper Chen et al. (2022). Subsequently, we apply our proposed method to compress these trained models. All experimental procedures are executed using PyTorch (Paszke et al., 2019) on NVIDIA RTX A6000 GPU units.

**Baselines.** We compare our compression paradigm with other baselines: The original NeRF model with MLP (Mildenhall et al., 2021), the uncompressed TensoRF-CP and TensoRF-VM from Chen et al. (2022), two prior compression methods for TensoRF-VM based on pruning and vector quantization named **VQ-TensoRF** from Li et al. (2023a) and **Re:TensoRF** from Deng & Tartaglione (2023).

**Hyperparameters.** As discussed in Section 3.1, our decoder has two transposed convolutional layers with SELU activation (Klambauer et al., 2017). They both have kernel size of 3, with stride 2 and padding 1. Thus, each layer has the upsample factor of 2. Given a feature plane sized $C_i \times W_i \times H_i$, we initialize the corresponding latent code $\mathbf{Z}_i$ to have the size of $C_{Z_i} \times W_i/4 \times H_i/4$.

Having a decoder with more parameters will enhance the model's decoding ability, while also increase its size. In light of this trade-off, we introduce two configurations: **TC-TensoRF-H** (stands for Transform Coding TensoRF - high compression) employs latent codes with 192 channels and a decoder with 96 hidden channels, while **TC-TensoRF-L** (low compression) has 384 latent channels and 192 decoder hidden channels. Regarding the hyperparameter $\lambda$, we experiment within the set $\{0.02, 0.01, 0.005, 0.001, 0.0005, 0.0002, 0.0001\}$, with higher $\lambda$ signifies a more compact model.

We train our models for $30,000$ iterations with Adam optimizer (Kingma & Ba, 2014). We use an initial learning rate of 0.02 for the latent codes and 0.001 for the networks, and apply an exponential learning rate decay.

## 4.2 EXPERIMENT RESULTS

**Quantitative Results.** We first compare our results with the baselines quantitatively. We use PSNR and SSIM (Wang et al., 2004) metrics to evaluate the reconstruction quality. The compression rate is determined by the compressed file size, represented in MB.

Our detailed results can be observed in Table 1. Compare to the other two TensoRF compression baselines VQ-TensoRF and Re:TensoRF, our variant TC-TensoRF-L showcases superior reconstruction performance in terms of both the PSNR and SSIM metrics while maintaining a reduced file size across 3 datasets: Synthetic-NeRF, Synthetic-NSVF and Tanks & Temples. In the case of the LLFF dataset, we are slightly behind VQ-TensoRF and Re:TensoRF in PSNR. Despite this, our SSIM surpass both baselines, and remarkably, the size of our compressed files is just about half of VQ-TensoRF and a mere quarter when compared to Re:TensoRF. For less number of channels, our TC-TensoRF-H is able to compress the model sizes to less than 2MB, while maintaining a decent reconstruction quality. Notably, our TC-TensoRF-H has similar SSIM with Re:TensoRF on Synthetic-NeRF and Tanks&Temples.

**Qualitative Results.** We showcase images rendered using our conpression method for both configurations: TC-TensoRF-L and TC-TensoRF-H in Figure 2 and Figure 3. Visually, there is minimal disparity between the uncompressed and compressed images.

**Rate-distortion performance.** The rate-distortion curve is widely used in neural compression to compare the compression performance across different compression level. Here we analyze the rate-distortion curve of our TC-TensoRF-L with various $\lambda$ values versus the baseline VQ-TensoRF with various codebook size. For the VQ-TensoRF evaluations, we employed the officially released code

Table 1: Quantitative results comparing our method versus the baselines. PSNR is measured in dB, while the sizes are in MB. We choose the $\lambda$ to balance between the reconstruction quality and storage size.

| | Methods | Synthetic-NeRF | | | Synthetic-NSVF | | | LLFF | | | Tanks and Temples | | |
|---|---|---|---|---|---|---|---|---|---|---|---|---|---|
| | | PSNR | SSIM | Size | PSNR | SSIM | Size | PSNR | SSIM | Size | PSNR | SSIM | Size |
| Uncompressed | NeRF | 31.01 | 0.947 | 5.0 | - | - | - | 26.50 | 0.811 | 5.0 | 25.78 | 0.864 | 5.0 |
| | TensoRF-CP | 31.56 | 0.949 | 3.9 | 34.48 | 0.971 | 3.9 | - | - | - | 27.59 | 0.897 | 3.9 |
| | TensoRF-VM | 33.09 | 0.963 | 67.6 | 36.72 | 0.982 | 71.6 | 26.70 | 0.836 | 179.8 | 28.54 | 0.921 | 72.6 |
| Compressed | VQ-TensoRF | 32.86 | 0.960 | 3.6 | 36.16 | 0.980 | 4.1 | 26.46 | 0.824 | 8.8 | 28.20 | 0.913 | 3.3 |
| | Re:TensoRF | 32.81 | 0.956 | 7.9 | 36.14 | 0.978 | 8.5 | 26.55 | 0.797 | 20.2 | 28.24 | 0.907 | 6.7 |
| | TC-TensoRF-L (ours) | 32.93 | 0.961 | 3.4 | 36.34 | 0.980 | 4.0 | 26.44 | 0.826 | 4.9 | 28.42 | 0.915 | 2.9 |
| | TC-TensoRF-H (ours) | 32.31 | 0.956 | 1.6 | 35.33 | 0.974 | 1.6 | 25.72 | 0.786 | 1.7 | 28.08 | 0.907 | 1.6 |

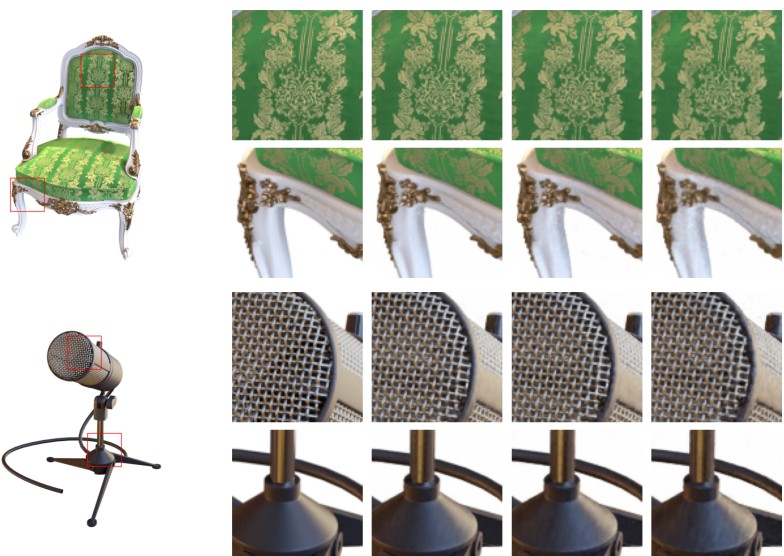

Figure 2: Qualitative results on Chair and Mic scenes from the Synthetic-NeRF dataset. From left to right: Ground truth image, uncompressed TensoRF, TC-TensoRF-L, TC-TensoRF-H.

and utilized the same pre-trained TensoRF models for consistency. Figure 4 shows that our TC-TensoRF-L outpaces VQ-TensoRF across various levels of compression in Synthetic-NeRF dataset with both PSNR and SSIM metrics. Rate-distortion curves for other datasets can be found in the Appendix.

**Training and rendering time.** Training our compression algorithm for a scene from the Synthetic-NeRF dataset is completed in approximately 40 minutes when using an NVIDIA A6000 GPU. In terms of rendering, our approach adds a negligible overhead of roughly 2 seconds for the decompression of parameters. Once decompressed, the rendering procedure is the same as TensoRF.

### 4.3 ABLATION STUDIES

We conduct experiments to verify our design choice. We test on the Synthetic-NeRF datasets, with our TC-TensoRF-L architecture.

**Training without Importance-Weighted Loss.** We examine the rate-distortion curves of TC-TensoRF-L, trained both with and without importance weight, as depicted in Figure 5. At an identical PSNR of 32.98 dB, employing importance weight in training our model helps reduce the file size from 4.59 MB to 3.92 MB.

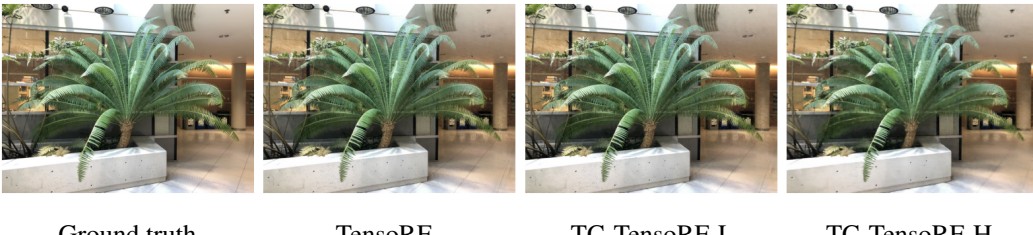

| Ground truth | TensoRF | TC-TensoRF-L | TC-TensoRF-H |

Figure 3: Qualitative results on Fern scene from the LLFF dataset.

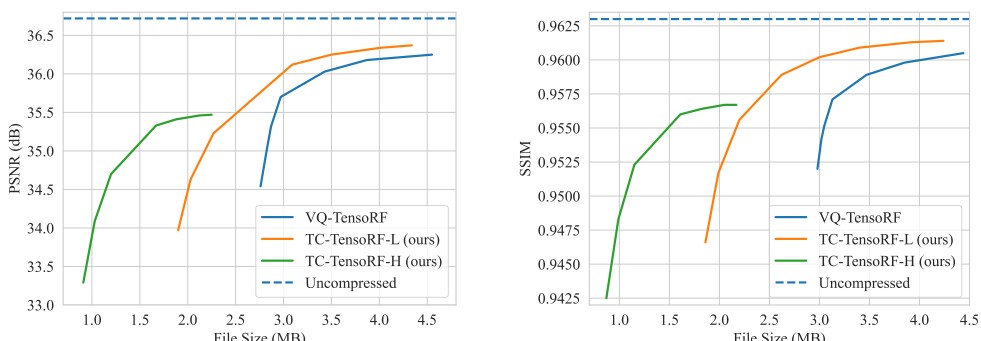

Figure 4: Comparison of rate-distortion curves between our proposed methods and the baseline VQ-TensoRF on the Synthetic-NeRF dataset. The left figure illustrates PSNR against file size, and the right figure showcases SSIM in relation to file size.

**Effective of Masked Entropy Model.** To demonstrate the efficacy of our masked entropy model, we undertook a comparative analysis between the compression performance of TC-TensoRF-L using the conventional factorized prior (Ballé et al., 2016; 2018) and our masked model. The results related to rate distortion curves can be found in the left plot of Figure 5.

It's noteworthy that, due to the additional overhead introduced by sending the masks, our results lag slightly behind the factorized prior in a low-rate setting. Yet, in medium to high-rate regimes, our prior emerges superior compared to the traditional factorized prior. To illustrate, for a PSNR value of 32.98 dB, the compressed file with the factorized prior occupies 4.26 MB. In contrast, our method employing the proposed masked entropy model results in a reduced file size of 3.92 MB.

To further understand the behavior of our masked entropy model, we visualize the masks learned for the Chair and Mic scene from Synthetic-NeRF dataset in the right side of Figure 5. We can observe that the masks resemble the rendering objects when viewed from different angles, and they inherently ignore the background. This behavior is similar to the pruning strategies employed in prior grid-based NeRF compression works (Li et al., 2023a;b; Deng & Tartaglione, 2023).

## 5 RELATED WORKS AND DISCUSSION

**Grid-based NeRF compression.** Since storage cost is a significant challenge of grid-based NeRF, several methods were proposed to solve this problem. Li et al. (2023a) introduces a three-stage approach, integrating voxel pruning and vector quantization (Gray, 1984) through a learnable codebook. Similarly, Re:NeRF (Deng & Tartaglione, 2023) employs voxel pruning, but adopts a strategy of sequentially removing and reintegrating parameters to prevent significant drop in performance. Meanwhile, Takikawa et al. (2022) adopts the codebook idea from Instant-NGP (Müller et al., 2022), but substitutes hash-based encoding with a learned mapping that associates grid positions to corresponding codebook indices. However this approach requires considerable training memory. Li et al. (2023b) applies downsampling to the voxels and employs a network to enhance render quality. Our

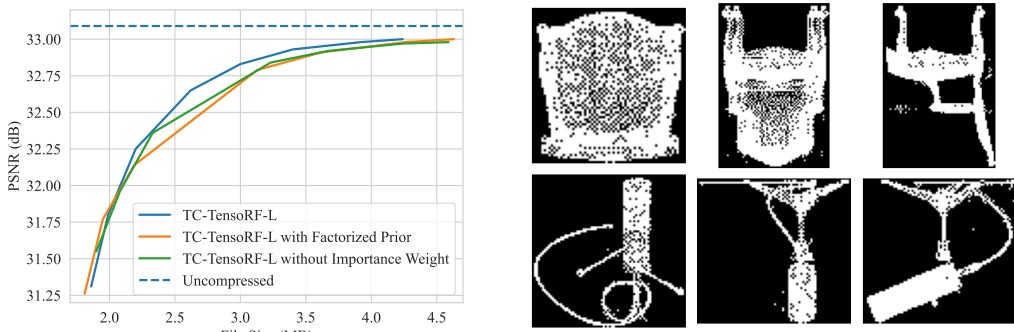

Figure 5: Ablation studies. Left: The rate-distortion curves are compared between different configurations. Right: Masks learned from Chair and Mic scenes in Synthetic-NeRF dataset.

method shares some similarity to Li et al. (2023b), but we learn the downsampled latent codes with a novel entropy model to effectively compress them. Additionally, while our masked factorized prior also resembles the pruning mechanism used in previous works, our method differentiates itself by adaptively learning the masks instead of relying on fixed thresholds.

**Neural compression for NeRF.**    Applying neural compression to NeRF is a relatively young field. Bird et al. (2021) learns an entropy model to compress the MLP-based NeRF Mildenhall et al. (2021) network weights, based on the prior model compression work of Oktay et al. (2019). In contrast, our work focuses on compressing the feature grids of grid-based NeRF. We additionally improve the conventional compression procedure and propose a novel entropy model.

**Discussion.**    Throughout this paper, our emphasis has been on applying neural compression techniques specifically to TensoRF. Nonetheless, our method has the potential to be applied to other grid-based NeRF methods beyond just TensoRF, such as Triplanes (Chan et al., 2022; Fridovich-Keil et al., 2023), DVGO (Sun et al., 2022) or Plenoxels (Fridovich-Keil et al., 2022). Taking DVGO as an example, we can learn a 4D latent code and have an entropy model to model its probability density. Then a decoder may decode this 4D latent code to render the scene. We hope to explore those extensions in future works.

## 6    CONCLUSION

In this study, we present a novel approach to applying neural compression to the TensoRF model, a prominent grid-based NeRF method. Our findings suggest that with the integration of our entropy model and compression techniques, we can achieve significant reductions in storage requirements with a small compromising in rendering quality. Although our primary focus is on the TensoRF model, the versatility of our approach means it has potential applications to other grid-based NeRF methods.

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

# A APPENDIX

## A.1 ALGORITHM

---
**Algorithm 1:** Transform Coding for TensoRF-VM

---
**Input** : Pretrained TensoRF-VM model
**Output:** Compressed TensoRF-VM model

Calculate $\{\mathbf{W}_i\}_{i=1}^3$ using Eq 4 and 5;
Initialize $\{\mathbf{Z}_i\}_{i=1}^3$ as 0-tensors;
Initialize decoder $D$ and entropy model $P$ with masks parameters $\{\pi_{\mathbf{M}_i}^0\}_{i=1}^3$ and $\{\pi_{\mathbf{M}_i}^1\}_{i=1}^3$;

**while** *not reached max_iterations* **do**
  Sample $\{\mathbf{M}_i\}_{i=1}^3$ using Gumbel-Softmax as in Eq 9;
  Reconstruct $\{\hat{\mathbf{P}}_i\}_{i=1}^3$ by Eq 8;
  Render the scene with $\{\hat{\mathbf{P}}_i\}_{i=1}^3$;
  Calculate the loss in Eq 10 and update the model;

**return** $(\{\mathbf{Z}_i\}_{i=1}^3, D, P)$ and other fine-tuned components.

---

## A.2 MORE RATE-DISTORTION COMPARISON

We further compare the rate-distortion curves of TC-TensoRF and the baseline VQ-TensoRF on the Synthetic-NSVF, LLFF and Tanks&Temples datasets.

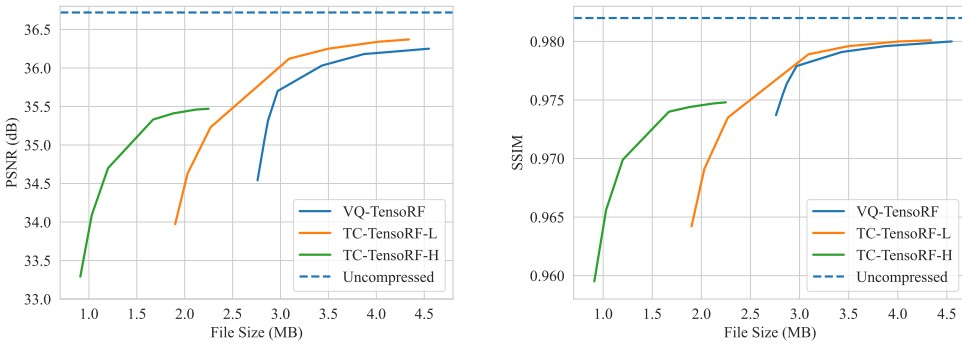

Figure 6: Comparison on Synthetic-NSVF dataset.

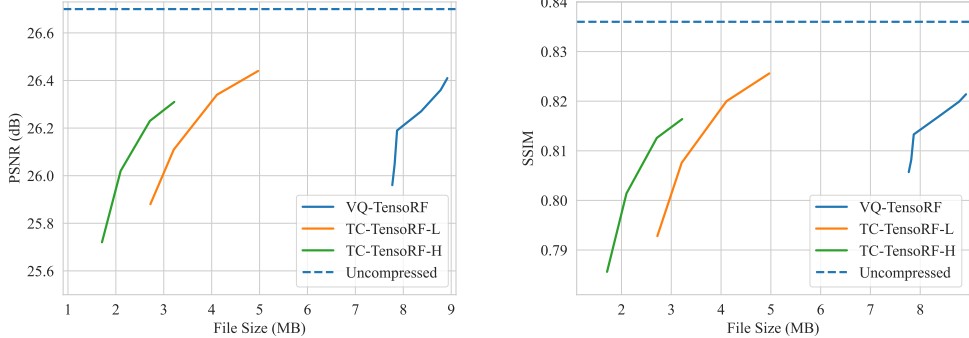

Figure 7: Comparison on LLFF dataset.

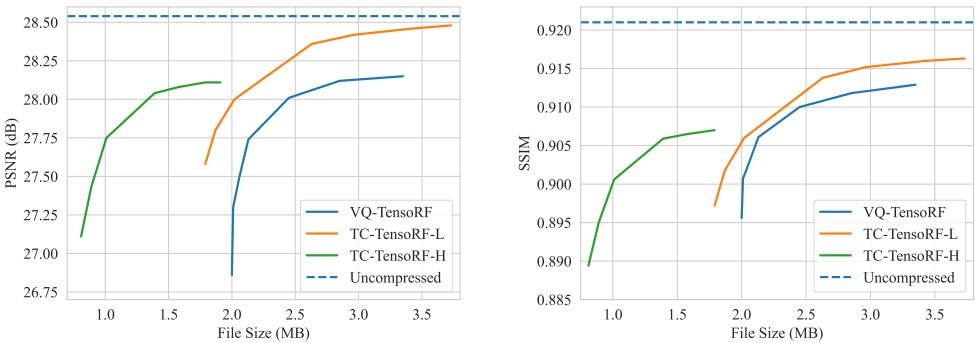

Figure 8: Comparison on Tanks and Temples dataset.

## A.3 MORE QUALITATIVE RESULTS

We show qualitative results on all scenes from Synthetic-NeRF, Synthetic-NSVF, LLFF and Tanks&Temples datasets.

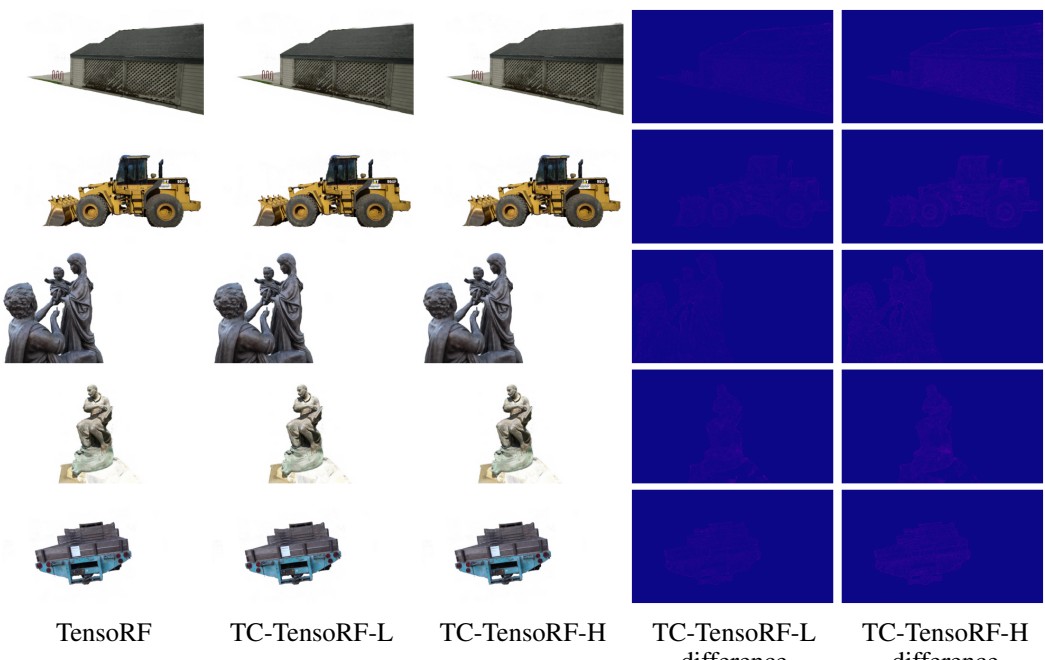

Figure 9: Qualitative results on Tanks and Temples dataset.

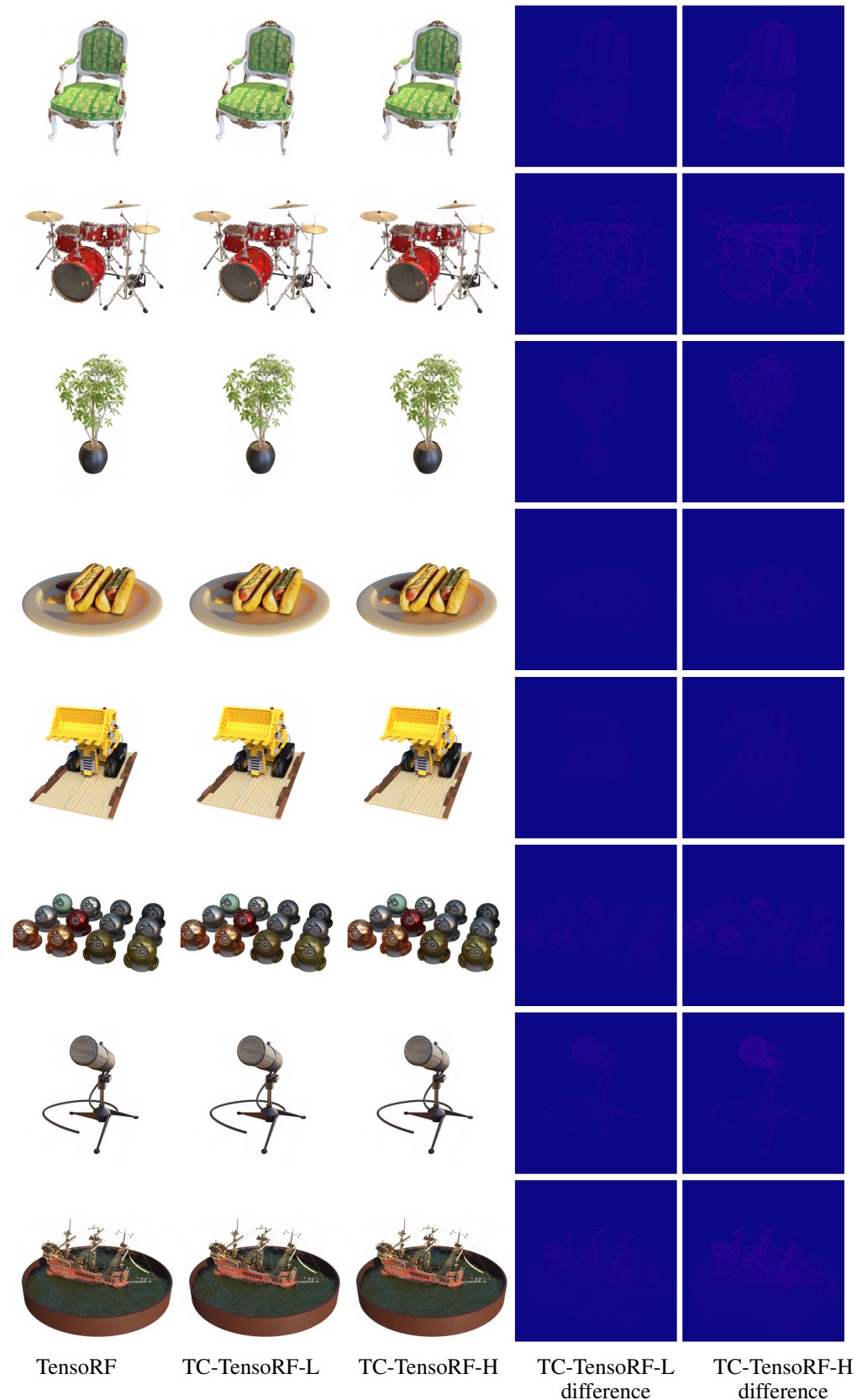

TensoRF     TC-TensoRF-L     TC-TensoRF-H     TC-TensoRF-L difference     TC-TensoRF-H difference

Figure 10: Qualitative results on Synthetic-NeRF dataset.

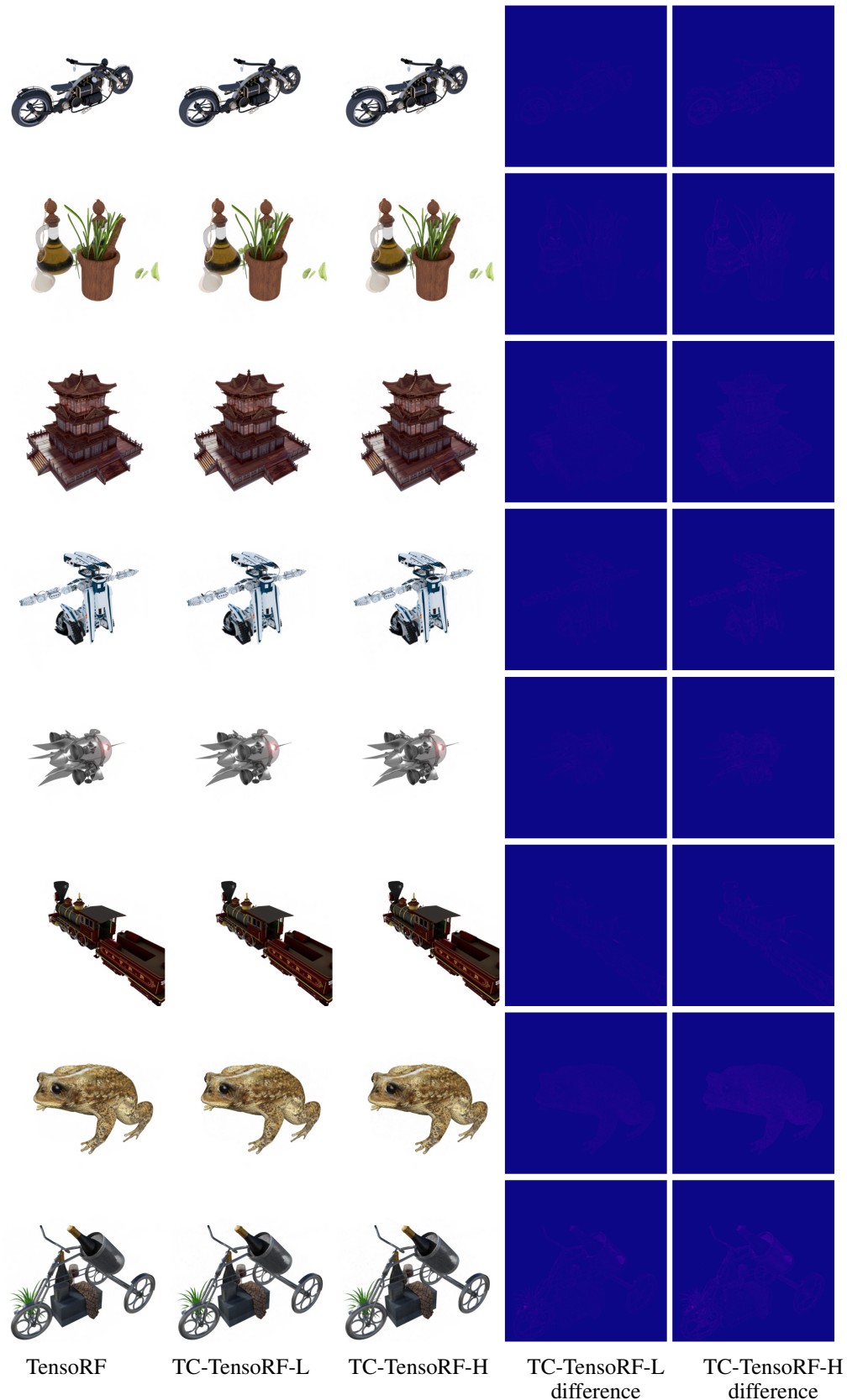

TensoRF    TC-TensoRF-L    TC-TensoRF-H    TC-TensoRF-L difference    TC-TensoRF-H difference

Figure 11: Qualitative results on Synthetic-NSVF dataset.

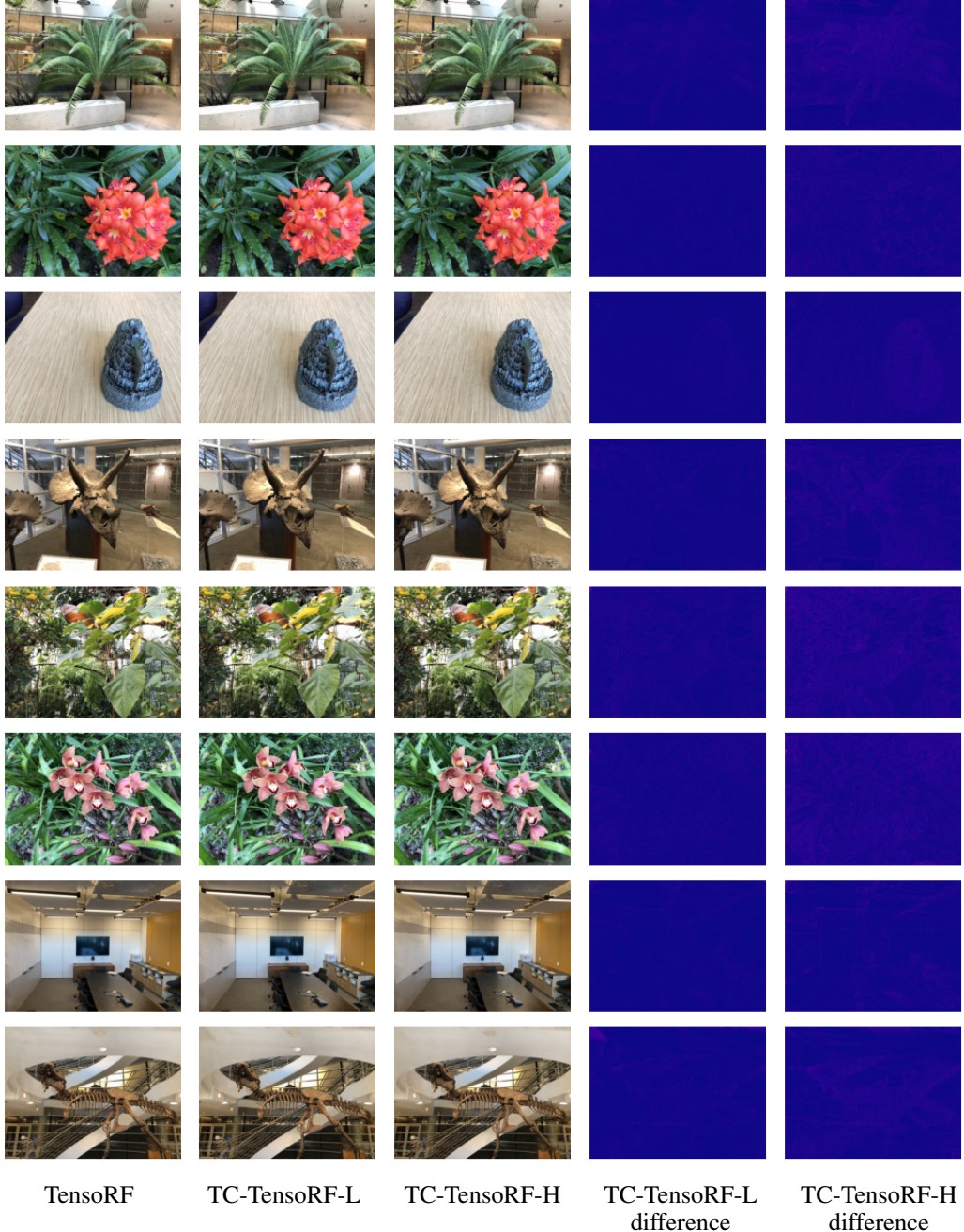

|  |  |  |  |  |
| TensoRF | TC-TensoRF-L | TC-TensoRF-H | TC-TensoRF-L difference | TC-TensoRF-H difference |

Figure 12: Qualitative results on LLFF dataset.

