# OpenReview forum: "NeRF Compression via Transform Coding"
_ICLR.cc/2024/Conference — Submitted to ICLR 2024_

### Official Review · Reviewer_L3yU · 2023-10-29

**Soundness:** 3 good
**Presentation:** 3 good
**Contribution:** 2 fair
**Rating:** 5
**Confidence:** 4

**Summary:**

This paper proposes a novel compression method for grid-based NeRF models, such as TensoRF. It optimized three latent codes and a decoder during training time, after training the full feature grid was omitted to reduce storage size. To achieve original rendering quality it directly reconstructs the full resolution feature plane with the learned latent code and decoder. Furthermore, an importance-weighted training loss was adopted to push the optimization focus on grid location that contributes more in rendering and a binary entropy mask to further reduce redundancy.

**Strengths:**

The semantics of the paper is smooth, and the proposed method is simple yet effective, it achieves better performance than previous methods that have a more sophisticated procedure, and the overall framework is very straightforward, so there should be no difficulty for other to reproduce.

**Weaknesses:**

1. The improvement over the previous method is not so significant,  the TC-TensoRF-L only shows notable improvement in the LLFF dataset, while in other datasets, it only brings minor performance gain in both visual quality（PSNR）and storage size. Though it provides a trade-off curve against VQ-TensoRF, the author only changed the codebook size of VQ-TensoRF, which may indicate that the curve was not drawn on the optimal hyper-parameter for VQ-TensoRF.

2.  Though the author has discussed DVGO and Plenoxels and tends to treat applying compression to those methods as a future work, I believe this could be a major weakness to not showing that the proposed is capable of generalize to other grid-based methods. As those methods have different design methodology e.g. plenoxels is already sparse and do not rely on MLP to recover feature in grid point. and it is important to show the proposed transform coding can still work for different types of grid-based nerf.

**Questions:**

what is the final composition of the storage size? It would be better to display the composition in a chart or figure to help others better understand the proposed method.

---

> ### Author Response · Authors · 2023-11-21
> **Response to Reviewer L3yU**
>
> We thank the reviewer for your constructive comments and for the recognition of our neural compression approach for NeRFs. We answer each of the reviewer's concerns below:
>
> **Response to the Weakness section:**
>
> > The improvement over the previous method is not so significant, the TC-TensoRF-L only shows notable improvement in the LLFF dataset, while in other datasets, it only brings minor performance gain in both visual quality（PSNR）and storage size. Though it provides a trade-off curve against VQ-TensoRF, the author only changed the codebook size of VQ-TensoRF, which may indicate that the curve was not drawn on the optimal hyper-parameter for VQ-TensoRF.
>
> In our experiments, our method demonstrates consistent improvements over VQ-TensoRF across four datasets. Given the high baseline performance of current methodologies and the maximum achievable quality of uncompressed models, we consider the enhancements in both reconstruction quality and storage efficiency achieved by our method to be substantial.
>
> In our rate-distortion curve experiments, the aim was to evaluate the performance of our method against VQ-TensoRF under various compression levels. To achieve this, we only adjusted the codebook size hyperparameter for VQ-TensoRF and the rate-distortion hyperparameter λ for our model. We believe our setting is fair for both methods. While additional tuning of hyperparameters might improve the compression capabilities of both models, we believe that the extensive effort required for such optimization is not feasible for practical, real-world use of NeRF compression.
>
> > Though the author has discussed DVGO and Plenoxels and tends to treat applying compression to those methods as a future work, I believe this could be a major weakness to not showing that the proposed is capable of generalize to other grid-based methods. As those methods have different design methodology e.g. plenoxels is already sparse and do not rely on MLP to recover feature in grid point. and it is important to show the proposed transform coding can still work for different types of grid-based nerf.
>
> We acknowledge that currently our method is specifically designed for feature plane-based NeRF, which may seem more restrictive compared to other methods. However, we believe that applying neural compression to NeRF is an interesting direction, and our work provides valuable insights and sets a foundation for future extensions of neural compression to other NeRF architectures.
>
> **Response to the questions:**
>
> > what is the final composition of the storage size? It would be better to display the composition in a chart or figure to help others better understand the proposed method.
>
> The average storage size breakdown of our model on the NeRF Synthetic dataset is provided in the following table. For the feature planes, we compressed it with the learned entropy model. All the other components (the renderer network, the decoder, density/appearance vectors, learned mask, entropy bottleneck parameters and model config) are packed into a single file and compressed with LZ77 using numpy.
>
> |Component|Storage Size in MB|
> |---|---|
> |TensoRF vector components|0.109|
> |TensoRF render networks|0.072|
> |TensoRF query mask|0.084|
> |Feature planes|1.657|
> |Decoder|1.380|
> |Entropy model parameters|0.020|
> |Entropy model masks|0.039|
> |Model config|0.042|
> |Total|3.403|

---

### Official Review · Reviewer_n1mE · 2023-10-30

**Soundness:** 3 good
**Presentation:** 3 good
**Contribution:** 2 fair
**Rating:** 5
**Confidence:** 4

**Summary:**

This paper proposes to compress the NeRF model’s feature grids using end-to-end optimized neural compression. By transmitting the latent code and a lightweight decoder, this method can significantly reduce the storage costs of NeRFs. Experiments on different
datasets show that this method is capable of compressing diverse NeRF scenes to a smaller size and outperforms previous works.

**Strengths:**

1) The paper is overall well-written and easy to read.

2) The proposed method applies a neural transform coding framework to compress the feature planes in NeRF. The overall performance is good, especially at relatively low bitrates. This method can be treated as a new route for NeRF compression besides network pruning and quantization.

**Weaknesses:**

1) Compared with existing work like VQ-TensorRF, the performance gain of the proposed method under a similar compression ratio is marginal in some cases (for low compression). Besides, this work is currently only designed for feature plane-based NeRF methods while other compared methods are more generalized.

2) The training and rendering time, especially the time needed for compressing and decompressing the features should be provided and compared with previous works.

3) The overall contribution of this paper is not sufficient enough. The transform coding framework is also similar to existing frameworks in image and video compression.

**Questions:**

The author proposed the latent decoder without an encoder. It will better demonstrate the efficiency by also showing the performance of a compressor with both an encoder and decoder during training and then drop the encoder during inference.

---

> ### Author Response · Authors · 2023-11-21
> **Response to Reviewer n1mE**
>
> We thank the reviewer for your constructive comments and for the recognition of our neural compression approach for NeRFs. We answer each of the reviewer's concerns below:
>
> **Response to the Weakness section:**
>
> > Compared with existing work like VQ-TensoRF, the performance gain of the proposed method under a similar compression ratio is marginal in some cases (for low compression). Besides, this work is currently only designed for feature plane-based NeRF methods while other compared methods are more generalized.
>
> Our method actually demonstrates consistent improvements over VQ-TensoRF across four datasets. Notably, in the LLFF dataset, our model achieves a similar level of reconstruction quality to VQ-TensoRF while requiring only half the model size. Additionally, our rate-distortion curve outperforms that of VQ-TensoRF.
>
> We acknowledge that currently our method is specifically designed for feature plane-based NeRF, which may seem more restrictive compared to other methods. However, we believe that applying neural compression to NeRF is an interesting direction, and our work provides valuable insights and sets a foundation for future extensions of neural compression to other NeRF architectures.
>
> > The training and rendering time, especially the time needed for compressing and decompressing the features should be provided and compared with previous works.
>
> For the training time comparison: Training our compression model on a scene from the NeRF-Synthetic dataset takes around 40 minutes on NVIDIA RTX A6000. The baseline VQ-TensoRF is faster and takes around 7 minutes on the same machine. For the baseline Re:TensoRF, we do not have the official code and can not measure the training time.
>
> At rendering time, our decompression procedure only takes 2 seconds more. VQ-TensoRF extraction takes around 0.02 second.
>
> > The overall contribution of this paper is not sufficient enough. The transform coding framework is also similar to existing frameworks in image and video compression.
>
> While our framework adapts the traditional transform coding methods used in image and video compression, it is distinct in our application of transform coding to NeRF. This application presents unique challenges and requirements, distinct from conventional images and videos. Our approach adapts and extends the principles of transform coding to address these specific challenges in 3D scene representation and compression. To further enhance the performance of our framework for NeRF, we proposed a unique framework with the omission of the encoder, a lightweight decoder; combined with an importance-weighted reconstruction loss and a masked entropy model. We believe our work also serves as a bridge, bringing well-established techniques from the realm of 2D media compression into the emerging field of 3D scene reconstruction. This not only provides a new perspective on familiar techniques but also opens up avenues for further research and development in NeRF compression.
>
> **Response to the questions:**
>
> > The author proposed the latent decoder without an encoder. It will better demonstrate the efficiency by also showing the performance of a compressor with both an encoder and decoder during training and then drop the encoder during inference.
>
> We show the comparison below for the NeRF-Synthetic dataset with our low compression architecture. Removing the encoder helped improve the compression performance. Note that this result is from one of our preliminary experiments, and we did not use the masked entropy model at that time.
>
> |λ Values|PSNR with encoder|Size with encoder|PSNR without encoder|Size without encoder|
> |---|---|---|---|---|
> |0.0001|32.90|4.67|33.00|4.63|
> |0.0002|32.88|4.33|32.98|4.26|
> |0.0005|32.85|3.83|32.91|3.62|
> |0.001|32.65|2.90|32.79|3.13|
> |0.005|32.15|2.35|32.14|2.19|
> |0.01|31.66|2.16|31.77|1.95|
> |0.02|30.88|2.02|31.26|1.81|

---

> > ### Comment · Reviewer_n1mE · 2023-11-22
> >
> > Thank you for your response. I will keep my rating.

---

### Official Review · Reviewer_GuYn · 2023-10-30

**Soundness:** 2 fair
**Presentation:** 3 good
**Contribution:** 2 fair
**Rating:** 3
**Confidence:** 4

**Summary:**

The paper proposes a novel method for efficiently compressing a grid-based NeRF model. It utilizes neural compression method to compress the model's feature grids. Specifically, the authors design an encoder-free architecture with a lightweight decoder, and present a weighted-rate-distortion loss incorporated a masked entropy coding mechanism to reduce redundance. The results show that the proposed method outperforms existing works.

**Strengths:**

1. The paper is well-written and effectively communicates the proposed method and its technical details. The authors provide clear explanations of the neural compression based framework.
2. The paper introduces a novel neural compression based framework for NeRF, which is a unique approach compared to existing compression methods that rely on pruning or vector quantization.

**Weaknesses:**

1. While the introduction of a neural compression method to encode the grid representation has demonstrated superior RD performance, the underlying motivation for this approach is not adequately elucidated

2. The novelty of this paper is not sufficient. The introduction of the weighted-rate-distortion loss lacks significant new contributions, as it heavily relies on an existing importance score calculation method [1]. The proposed binary mask entropy coding resembles the concept of importance map-based bit allocation schemes, such as in [2]. Moreover, the compression method falls short in considering contextual information for further reduction of spatial redundancy.

[1] Compressing Volumetric Radiance Fields to 1 MB. CVPR 2023.
[2] Learning End-to-End Lossy Image Compression: A Benchmark. TPAMI 2022.

3. It would be beneficial to include a comparison of the training time between the proposed method and other established methods. Additionally, the paper should provide a more precise breakdown of the storage sizes for each component, including the decoder, entropy model, binary mask, and grid feature.

4. The term 'transform coding' in the paper's title may not be entirely appropriate, as the paper employs an encoder-free framework that directly learns the latent code without involving analysis transforms within the coding paradigm

**Questions:**

1."What are the advantages of the proposed compression method when compared to existing pruning or vector quantization methods?"
2."Have the authors considered the results of incorporating an encoder and training the entire network end-to-end?"
3."It is recommended that the authors provide a detailed breakdown of the storage size for each component to offer a more comprehensive understanding."
4. In Figure 5, I noticed that the blue curve (TC-TensoRF-L) is situated below the orange (w/ Factorized Prior) and green (w/o Importance Weight) curves in the lower bitrate range. Could you please provide an explanation for this observation?

---

> ### Author Response · Authors · 2023-11-21
> **Response to Reviewer GuYn**
>
> We thank the reviewer for your constructive comments and for the recognition of our unique approach compared to existing compression methods. We answer each of the reviewer's concerns below:
>
> **Response to the Weakness section:**
>
> >While the introduction of a neural compression method to encode the grid representation has demonstrated superior RD performance, the underlying motivation for this approach is not adequately elucidated.
>
> Our underlying motivation for using neural compression is grounded in its proven efficacy in image compression, where it has consistently demonstrated superior rate-distortion (R-D) performance compared to traditional compression methods. Its ability to efficiently manage high-dimensional data make it particularly suitable for handling the complexities associated with grid representations in NeRFs. Upon examination of grid-based methods such as TensoRF, we observed that the grid features often exhibit significant redundancies and can be effectively compressed with neural compression.
>
> > The novelty of this paper is not sufficient. The introduction of the weighted-rate-distortion loss lacks significant new contributions, as it heavily relies on an existing importance score calculation method [1]. The proposed binary mask entropy coding resembles the concept of importance map-based bit allocation schemes, such as in [2]. Moreover, the compression method falls short in considering contextual information for further reduction of spatial redundancy.
>
> In response to the concerns of The Reviewer:
>
> 1. Adaptation of Importance Weighted Loss: We agree that our approach is based on an established importance score calculation method. However, our unique application of this score as a training weight in our model represents a contribution. Unlike previous works [1] that applied this score with a pruning threshold and required hyperparameters tuning, our method integrates this score with the rate-distortion loss commonly used in neural compression. This integration enables our model to adeptly prioritize critical grid locations across varying compression levels λ.
>
> 2. Binary Mask Entropy Coding in NeRFs: While there are conceptual similarities with existing importance map-based bit allocation techniques, our application in the realm of NeRFs is notably distinct. Our strategy leverages the sparsity characteristic of NeRF representations to optimize the encoding process, setting our approach apart from traditional methods used for images. Additionally, our use of Gumbel-Softmax for automatic learning of the mask is a unique aspect that differentiates our method from prior works.
>
> 3. Contextual Information for Spatial Redundancy Reduction: We acknowledge the point raised about the potential for further reducing spatial redundancy through contextual information. This presents an opportunity for future work and improvement.
>
> 4. In our paper, we also proposed several modifications to the existing neural compression framework. These include the omission of the encoder and the implementation of a light-weighted decoder. While we recognize that these modifications may not be groundbreaking in their novelty, we believe that they play a crucial role in enhancing the overall efficiency and effectiveness of our framework.

---

> ### Author Response · Authors · 2023-11-21
> **Response to Reviewer GuYn (cont.)**
>
> > It would be beneficial to include a comparison of the training time between the proposed method and other established methods. Additionally, the paper should provide a more precise breakdown of the storage sizes for each component, including the decoder, entropy model, binary mask, and grid feature.
>
> For the training time comparison: Training our compression model on a scene from the NeRF-Synthetic dataset takes around 40 minutes on NVIDIA RTX A6000. The baseline VQ-TensoRF is faster and takes around 7 minutes on the same machine. For the baseline Re:TensoRF, we do not have the official code and can not measure the training time.
>
> The average storage size breakdown of our model on the NeRF Synthetic dataset is provided in the following table. For the feature planes, we compressed it with the learned entropy model. All the other components (the renderer network, the decoder, density/appearance vectors, learned mask, entropy bottleneck parameters and model config) are packed into a single file and compressed with LZ77 using numpy.
>
> |Component|Storage Size in MB|
> |---|---|
> |TensoRF vector components|0.109|
> |TensoRF render networks|0.072|
> |TensoRF query mask|0.084|
> |Feature planes|1.657|
> |Decoder|1.380|
> |Entropy model parameters|0.020|
> |Entropy model masks|0.039|
> |Model config|0.042|
> |Total|3.403|
>
> > The term 'transform coding' in the paper's title may not be entirely appropriate, as the paper employs an encoder-free framework that directly learns the latent code without involving analysis transforms within the coding paradigm
>
> The term 'transform coding' in our title was selected to broadly encompass the conceptual essence of our approach, which involves transforming data into a more compressible format. While our paper indeed introduces an encoder-free framework that directly learns the latent code, we believe this process aligns with the fundamental principles of transform coding. The direct learning of the latent code can be viewed as an implicit form of analysis transform, where the input data is converted into a representation that is more amenable to efficient compression.
>
> However, we acknowledge that the term 'transform coding' might traditionally be associated with specific technical implementations involving explicit analysis and synthesis transforms. In light of your feedback, we are open to considering alternative names that may more precisely communicate the specifics of our methodology.
>
>
> **Response to the questions:**
>
> > What are the advantages of the proposed compression method when compared to existing pruning or vector quantization methods?
>
> Neural compression is a well-established method for compressing data with a strong theoretical foundation, and can yield state-of-the-art performance on images and videos [3]. On the other hand, we believe that vector quantization has some disadvantages, such as the need of a regularization to fully utilize the codebook [1]. We also empirically found that the neural compression works better than vector quantization and pruning in our experiments. Understanding the fundamental differences between the two frameworks in compression could be an interesting future work.
>
>
> > Have the authors considered the results of incorporating an encoder and training the entire network end-to-end?
>
> We considered training with an encoder. We show the comparison below for the NeRF-Synthetic dataset with our low compression architecture. Removing the encoder helped improve the compression performance. Note that this result is from one of our preliminary experiments, and we did not use the masked entropy model at that time.
>
> |λ Values|PSNR with encoder|Size with encoder|PSNR without encoder|Size without encoder|
> |---|---|---|---|---|
> |0.0001|32.90|4.67|33.00|4.63|
> |0.0002|32.88|4.33|32.98|4.26|
> |0.0005|32.85|3.83|32.91|3.62|
> |0.001|32.65|2.90|32.79|3.13|
> |0.005|32.15|2.35|32.14|2.19|
> |0.01|31.66|2.16|31.77|1.95|
> |0.02|30.88|2.02|31.26|1.81|
>
> > It is recommended that the authors provide a detailed breakdown of the storage size for each component to offer a more comprehensive understanding.
>
> We provided the breakdown of the storage size in our response to the Weakness section.
>
> > In Figure 5, I noticed that the blue curve (TC-TensoRF-L) is situated below the orange (w/ Factorized Prior) and green (w/o Importance Weight) curves in the lower bitrate range. Could you please provide an explanation for this observation?
>
> In the case of the Factorized Prior model, we hypothesize that at lower bitrate, the cost of sending the masks is more than the feature planes size saved when using the masked entropy model. Whereas in the comparison with the model not using the importance weight mechanism, at lower bitrate, TC-TensoRF-L is forced to completely ignore locations with low to medium importance, leading to lower reconstruction quality overall.

---

> ### Author Response · Authors · 2023-11-21
> **Response to Reviewer GuYn (cont.)**
>
> References:
>
> [1] Compressing Volumetric Radiance Fields to 1 MB. CVPR 2023.
>
> [2] Learning End-to-End Lossy Image Compression: A Benchmark. TPAMI 2022.
>
> [3] An Introduction to Neural Data Compression.  Foundations and Trends in Computer Graphics and Vision.

---

### Author Response · Authors · 2023-11-21
**Summary for our responses**

We thank the reviewers for their constructive comments. We want to highlight some keypoints in our response:

**Novelty concern**

While our framework adapts the traditional transform coding and neural compression methods used in image and video compression, it is distinct in our application of transform coding to NeRF. This application presents unique challenges and requirements, distinct from conventional images and videos. Our approach adapts and extends the principles of transform coding to address these specific challenges in 3D scene representation and compression. To further enhance the performance of our framework for NeRF, we proposed a unique framework with the omission of the encoder, a lightweight decoder; combined with an importance-weighted reconstruction loss and a masked entropy model. We believe our work also serves as a bridge, bringing well-established techniques from the realm of 2D media compression into the emerging field of 3D scene reconstruction. This not only provides a new perspective on familiar techniques but also opens up avenues for further research and development in NeRF compression.

**NeRF models limitation**

We acknowledge that currently our method is specifically designed for feature plane-based NeRF (such as TensoRF[1] or k-planes[2]), which may seem restrictive. However, we believe that applying neural compression to NeRFs is an interesting direction, and our work provides valuable insights and sets a foundation for future extensions of neural compression to other NeRF architectures.

References:

[1] TensoRF: Tensorial Radiance Fields. ECCV 2022

[2] K-Planes: Explicit Radiance Fields in Space, Time, and Appearance. CVPR 2023

---

### Meta-Review · Area_Chair_qVSk · 2023-12-11

**Metareview:**

This paper explores non-linear transform coding methods to compress the explicit grid NeRF representation efficiently. Experimental results with ablation studies show the effectiveness of the proposed method.

The major concerns of reviewers include limited novelty, insufficient contributions, and limited performance improvement.

Based on the provided rebuttal and the comments of reviewers, the paper is not ready for ICLR 2024.

**Justification For Why Not Higher Score:**

N/A

**Justification For Why Not Lower Score:**

N/A

---

### Decision · Program_Chairs · 2024-01-16

Reject